# Serologic and molecular evidence for circulation of Crimean-Congo hemorrhagic fever virus in ticks and cattle in Zambia

Masahiro Kajihara[1], Martin Simuunza[2,3], Ngonda Saasa[2], George Dautu[4], Akina Mori-Kajihara[1], Yongjin Qiu[1], Ryo Nakao[5], Yoshiki Eto[1], Hayato Furumoto[6], Bernard M. Hang'ombe[2,3], Yasuko Orba[1], Hirofumi Sawa[1,2], Edgar Simulundu[2], Shuetsu Fukushi[7], Shigeru Morikawa[8¤a], Masayuki Saijo[7], Jiro Arikawa[9], Swithine Kabilika[10], Mwaka Monze[11], Victor Mukonka[12], Aaron Mweene[2†], Ayato Takada[1,2]*, Kumiko Yoshimatsu[9¤b]*

1 Research Center for Zoonosis Control, Hokkaido University, Sapporo, Japan, 2 School of Veterinary Medicine, the University of Zambia, Lusaka, Zambia, 3 Africa Centre of Excellence for Infectious Diseases of Humans and Animals, University of Zambia, Lusaka, Zambia, 4 Central Veterinary Research Institute, Ministry of Fisheries and Livestock, Lusaka, Zambia, 5 Graduate School of Infectious Diseases, Faculty of Veterinary Medicine, Hokkaido University, Sapporo, Japan, 6 JICA Zambia Office, Japan International Cooperation Agency, Lusaka, Zambia, 7 Department of Virology I, National Institute of Infectious Diseases, Tokyo, Japan, 8 Department of Veterinary Science, National Institute of Infectious Diseases, Tokyo, Japan, 9 Department of Microbiology, Graduate School of Medicine, Hokkaido University, Sapporo, Japan, 10 Department of Veterinary Services, Ministry of Fisheries and Livestock, Lusaka, Zambia, 11 Virology Laboratory, University Teaching Hospital, Lusaka, Zambia, 12 Zambia National Public Health Institute, Lusaka, Zambia

† Deceased.
¤a Current address: Department of Microbiology, Faculty of Veterinary Medicine, Okayama University of Science, Imabari, Japan
¤b Current address: Institute for Genetic Medicine, Hokkaido University, Sapporo, Japan
* atakada@czc.hokudai.ac.jp (AT); yosimatu@igm.hokudai.ac.jp (KY)

**Data Availability Statement:** All sequence data are available from NCBI database https://www.ncbi.nlm.nih.gov/search/all/?term=LC534898 and

## Abstract

Crimean-Congo hemorrhagic fever (CCHF) is a tick-borne zoonosis with a high case fatality rate in humans. Although the disease is widely found in Africa, Europe, and Asia, the distribution and genetic diversity of CCHF virus (CCHFV) are poorly understood in African countries. To assess the risks of CCHF in Zambia, where CCHF has never been reported, epidemiologic studies in cattle and ticks were conducted. Through an indirect immunofluorescence assay, CCHFV nucleoprotein-specific serum IgG was detected in 8.4% (88/1,047) of cattle. Among 290 *Hyalomma* ticks, the principal vector of CCHFV, the viral genome was detected in 11 ticks. Phylogenetic analyses of the CCHFV S and M genome segments revealed that one of the detected viruses was a genetic reassortant between African and Asian strains. This study provides compelling evidence for the presence of CCHFV in Zambia and its transmission to vertebrate hosts.

https://www.ncbi.nlm.nih.gov/search/all/?term=LC534910.

**Funding:** This work was supported by the Agency for Medical Research and Development and Japan International Cooperation Agency within the framework of the Science and Technology Research Partnership for Sustainable Development (grant no. JP18jm0110019 to AT) and by the Agency for Medical Research and Development within the framework of the Japan Initiative for Global Research Network on Infectious Diseases (grant no. JP18fm0108008 to HS). Funding was provided in part by the Japan Society for the Promotion of Science (KAKENHI) (grant nos. JP16H02627 to AT, JP15K18778 to MK, and JP18K15163 to MK). The funders had no role in the study design, data collection and analysis, decision to publish, or preparation of the manuscript.

**Competing interests:** The authors have declared that no competing interests exist. Author Aaron Mweene was unable to confirm their authorship contributions. On their behalf, the corresponding author has reported their contributions to the best of their knowledge.

## Author summary

Crimean-Congo hemorrhagic fever (CCHF) is a severe viral disease mainly transmitted by ticks. Effective prophylactics and therapeutics have not been established for this disease yet. While CCHF is endemic in Africa, information on the distribution and genetic diversity of CCHF virus (CCHFV) is quite limited in many Sub-Saharan African countries. In this study, we conducted serologic and molecular epidemiologic investigations for CCHFV infection in cattle and ticks in Zambia. Serologic screening revealed that 8.4% of cattle were tested positive for CCHFV-specific IgG. *Hyalomma* ticks infected with CCHFV were also identified by genetic screening. Phylogenetic analyses showed that one of the CCHFVs detected in Zambia was a genetic reassortant between African and Asian CCHFV strains. Currently, Zambia is considered CCHF-free country because CCHF cases have never been reported. However, the findings in this study indicate that CCHFV is maintained in *Hyalomma* ticks and occasionally transmitted to vertebrate hosts such as cattle in Zambia. Further epidemiologic studies and continuous monitoring of CCHFV infection should be implemented in the southern African region.

## Introduction

Crimean-Congo hemorrhagic fever (CCHF) is a tick-borne zoonotic disease characterized by hemorrhagic fever and a high case fatality rate. CCHF virus (CCHFV) belongs to the family *Nairoviridae*, genus *Orthonairovirus* [1], and has a negative-sense and single-stranded RNA genome composed of tripartite large (L), medium (M), and small (S) segments encoding RNA-dependent RNA polymerase, glycoprotein, and nucleoprotein (N), respectively. Although CCHFVs have been detected in various tick species, *Hyalomma* ticks are the principal vector and reservoir of CCHFV [2]. A variety of wild and domestic animals, including cattle, goats, and sheep, are susceptible to the virus [2]. Generally, these animals do not manifest clinical symptoms upon CCHFV infection and serve as amplifying hosts of the virus. Therefore, direct contact with blood or tissues of infected livestock is a major transmission mode of CCHFV to humans, as well as tick bites. Nosocomial CCHFV infection in healthcare workers is also seen during CCHF outbreaks [3].

CCHFV is widely found across Africa, Europe, and Asia and has caused more than 1,000 annual cases in the past decade [4]. However, the epidemiology of CCHF in Sub-Saharan Africa remains poorly understood. Because other febrile diseases, such as malaria, are prevalently endemic in the region [5], sporadic or subclinical CCHFV infections have rarely been recognized. Therefore, despite the public health importance, viral hemorrhagic fevers, including CCHF, tend to be neglected until large-scale outbreaks attract public attention. For example, Zambia is currently categorized as a CCHF nonendemic country due to the absence of reported CCHF cases [6]. However, because *Hyalomma* ticks are commonly seen in Zambia and Zambia is surrounded by the countries where CCHF cases have been reported [2, 7], such as the Democratic Republic of the Congo, Namibia, Tanzania, and Zimbabwe, it is highly likely that CCHFV exists in Zambia.

In this paper, we carried out epidemiologic studies in cattle and *Hyalomma* ticks in Zambia to evaluate the risk of CCHF. Serologic screening identified anti-CCHFV antibody-positive cattle, and CCHFV genomes were also detected in adult *Hyalomma* ticks. The present study convincingly demonstrates the presence of CCHFV in Zambia and highlights the necessity of further epidemiologic studies on CCHFV infection of humans and animals in currently believed nonendemic countries, such as Zambia.

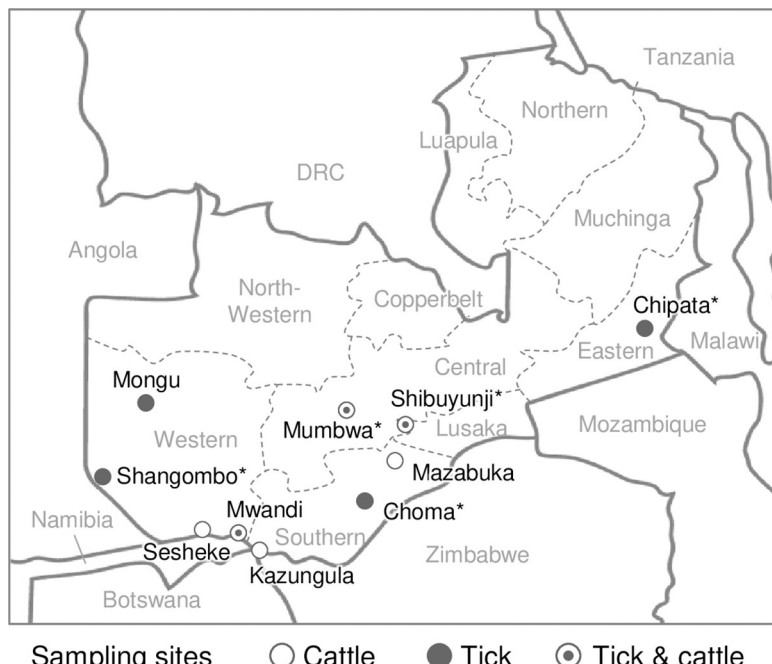

**Fig 1. Sampling sites in Zambia.** Cattle serum and *Hyalomma* tick collection sites are shown by white and black circles, respectively. The sites where both cattle and tick samples were collected are shown by black dots in white circles. Broken lines indicate provincial borders. DRC: the Democratic Republic of the Congo. *: Districts where the CCHFV genome was detected in ticks. The base layer of the map was downloaded from MapChart (https://mapchart. net/index.html).

## Methods

### Ethics statement

The present study was conducted as a collaborative study with Central Veterinary Research Institute, Ministry of Fisheries and Livestock, Zambia. Sample collection was approved by the Department of Veterinary Services according to the Animal Health Act No. 27 of 2010.

### Serum samples

A total of 1,047 individual cattle serum samples collected in the previous study [8] during 2012–2014 and a follow-up study in 2015 from traditional cattle herds in six districts of four provinces of Zambia (Shibuyunji in Lusaka Province, Mumbwa in Central Province, Mazabuka and Kazungula in Southern Province, and Mwandi and Sesheke in Western Province) were used for the serologic screening (Fig 1 and Table 1). For the genetic screening of CCHFV

**Table 1. Summary of serologic screening of cattle for Crimean-Congo hemorrhagic fever virus-specific IgG.**

| District | No. serum samples | No. (%) positive |
| --- | --- | --- |
| Kazungula | 150 | 3 (2.0) |
| Mazabuka | 111 | 6 (5.4) |
| Mumbwa | 131 | 23 (17.6) |
| Mwandi | 400 | 23 (5.8) |
| Sesheke | 150 | 22 (14.7) |
| Shibuyunji | 105 | 11 (10.5) |
| Total | 1,047 | 88 (8.4) |

genome, total RNA was extracted from randomly selected 526 samples out of 1,047 cattle sera using QIAamp Viral RNA Mini Kit (Qiagen) according to the manufacturer's instructions.

## Indirect immunofluorescence assay (IFA)

An IFA was performed as described previously [9]. Briefly, HeLa cells constitutively expressing N of CCHFV strain 8402 (accession no. AF403737) were mixed with parental HeLa cells at a ratio of 1:3 and washed with phosphate-buffered saline (PBS). The cells were then spotted onto 14-well glass slides, air dried, and fixed with acetone for 5 min. The slides were stored at –80˚C until use. A positive control (recombinant N-immunized rabbit serum [10]) and tested cattle sera were diluted at 1:160 with PBS. After applying serum samples followed by 1 h incubation at room temperature, the slides were washed 3 times with PBS. Alexa Fluor 488-conjugated goat anti–rabbit IgG (Thermo Fisher Scientific) and FITC-conjugated goat anti–bovine IgG (Kirkegaard & Perry Laboratories, Inc.) were used as secondary antibodies at 1:1,000 dilutions with PBS. After incubation for 1 h with secondary antibodies, the slides were washed as described above. Finally, each slide was covered with 50% glycerol in PBS and observed under a fluorescence microscope. Cattle sera showing clear immunofluorescence in the cytoplasm of approximately 25% of HeLa cells were regarded as positive. Serum samples showing no fluorescent signals or showing nonspecific signals almost all the cells were regarded as negative. Microscopic examination was carried out by two to four examiners. Serum samples for which judgment varied among the examiners were regarded as negative.

## Tick samples

A total of 290 adult *Hyalomma* ticks were collected from infested local cattle in 7 districts of 5 provinces of Zambia: Choma in Southern Province; Chipata in Eastern Province; Mongu, Mwandi, and Sesheke in Western Province; Mumbwa in Central Province; and Shibuyunji in Lusaka Province (Fig 1 and Tables 2 and 3). Tick species were identified morphologically using standard keys under a stereomicroscope (*Hyalomma marginatum*, n = 25; *H. truncatum*, n = 259; *Hyalomma* spp., n = 6) [7]. Ticks were individually washed with 70% ethanol and sterile PBS twice and then homogenized in 100 μL of plain Dulbecco's modified Eagle medium

**Table 2. Summary of genetic screening of *Hyalomma* ticks for Crimean-Congo hemorrhagic fever virus.**

| District | Tick species | No. tick samples | No. (%) positive |
|---|---|---|---|
| Choma | *H. marginatum* | 10 | 1 (10.0) |
| | *H. truncatum* | 22 | 1 (4.5) |
| | *Hyalomma* spp. | 5 | 0 (0) |
| Chipata | *H. truncatum* | 52 | 4 (7.7) |
| Mongu | *H. marginatum* | 5 | 0 (0) |
| | *H. truncatum* | 1 | 0 (0) |
| Mumbwa | *H. truncatum* | 110 | 2 (1.8) |
| Mwandi | *H. marginatum* | 9 | 0 (0) |
| | *H. truncatum* | 3 | 0 (0) |
| | *Hyalomma* sp. | 1 | 0 (0) |
| Shangombo | *H. marginatum* | 1 | 0 (0) |
| | *H. truncatum* | 41 | 1 (2.4) |
| Shibuyunji | *H. truncatum* | 30 | 2 (6.7) |
| Total | | 290 | 11 (3.8) |

**Table 3. Summary of *Hyalomma* ticks displayed by species, sex, and genetic screening results.**

| Tick | Sex | No. tick samples | No. (%) positive |
|---|---|---|---|
| *H. marginatum* | M | 13 | 0 (0) |
| | F | 12 | 1 (8.3) |
| | Subtotal | 25 | 1 (4.0) |
| *H. truncatum* | M | 165 | 8 (4.8) |
| | F | 94 | 2 (2.1) |
| | Subtotal | 259 | 10 (3.9) |
| *Hyalomma* spp. | M | 0 | ND* |
| | F | 6 | 0 (0) |
| | Subtotal | 6 | 0 (0) |
| Total | | 290 | 11 (3.8) |

* ND: not determined.

(Nissui) by using a Micro Smash MS100R (TOMY) for 30 s at 3,000 rpm as described previously [11]. Total RNA was extracted from tick homogenates using TRIzol reagent (Invitrogen) according to the manufacturer's instructions.

## Virus isolation

As a routine screening of tick-borne viruses, tenfold-diluted tick homogenates were filtrated through 0.45 μm membrane filters (Iwaki) and then inoculated onto Vero E6 (African green monkey kidney) cells. Cells were maintained in Dulbecco's modified Eagle medium supplemented with 2% fetal bovine serum, 2 mM L-glutamine, 4% antibiotic–antimycotic solution (Gibco), and 1.0 mg/mL $NaHCO_3$ for 14 days. Blind passages were performed at 14 days postinfection. RNAs were purified from supernatants using the QIAamp viral RNA minikit (Qiagen), and virus growth was examined by reverse transcription PCR (RT-PCR). These experiments were performed in the Biosafety level 3 laboratory in Hokudai Center for Zoonosis Control in Zambia, the School of Veterinary Medicine, the University of Zambia.

## Genetic screening of tick samples for CCHFV

Tick and cattle total RNAs were screened for the CCHFV S genome segment by nested RT-PCR following a previous report [12]. First-round one-step RT-PCR was performed using a QIAGEN OneStep RT-PCR Kit (QIAGEN) with the primer set CCHF-F2 (5'-TGGACACC TTCACAAACTC-3') and CCHF-R3 (5'-GACAAATTCCCTGCACCA-3'). The one-step RT-PCR program consisted of reverse transcription at 50°C for 30 min; initial PCR activation at 95°C for 15 min; followed by 35 cycles of denaturation at 94°C for 30 s, annealing at 52°C for 30 s, and extension at 72°C for 30 s; and final extension at 72°C for 7 min (Veriti 200 thermal cycler; Life Technologies). The PCR products were subsequently subjected to secondround PCR using *TaKaRa Ex Taq* Hot Start Version (Takara) with the primer set CCHF-F3 (5'-GAGTGTGCCTGGGTTAGCTC-3') and CCHF-R2 (5'-GACATTACAATTTCGCCA GG-3'). The PCR program consisted of initial PCR activation at 98°C for 2 min; followed by 30 cycles of denaturation at 98°C for 10 s, annealing at 52°C for 30 s, and extension at 72°C for 30 s; and final extension at 72°C for 5 min. Genome amplification was visualized by electrophoresis in 2.0% agarose gels and ethidium bromide staining. CCHF genome detection was confirmed by repeated experiments.

## Statistical analysis

Association between CCHFV prevalence and tick species was analyzed by chi-square test using R version 4.0.5.

## Genetic analyses of CCHFVs

Complimentary DNA of the CCHFV S segment was synthesized with the primer CCHFV S 57F (5'-AATGGARAAYAARATHGARA-3') using SuperScript IV Reverse Transcriptase (Invitrogen) according to the manufacturer's instructions. Subsequently, approximately 1,300 nt of the S genome sequence was amplified with the primer set CCHFV-F2 and CCHFV S 1492R (5'-CRCTDGTRGCRTTVCCYTTRAC-3') using KOD FX Neo (TOYOBO). The PCR program consisted of initial PCR activation at 94˚C for 2 min; followed by 45 cycles of denaturation at 98˚C for 10 s, annealing at 50˚C for 30 s, and extension at 68˚C for 1.5 min; and final extension at 68˚C for 6 min. Nested RT-PCR was performed to amplify two separated regions of the M segment according to a previous report [13]. Nucleotide sequences of amplified products were determined by Sanger sequencing using a BigDye Terminator v3.2 Cycle Sequencing Kit (Thermo Fisher Scientific) and a 3130 Genetic Analyzer (Applied Biosystems). The nucleotide sequences of the S and M segments were aligned together with the sequences of other CCHFV strains available from GenBank by using the built-in MUSCLE program in Molecular Evolutionary Genetics Analysis version 7 [14]. A total of 1,293 nt of the S segment and 535 and 515 nt of the 5' and 3' regions of the M segment, respectively, were used for subsequent phylogenetic analyses. The evolutionary relationship was inferred by using the maximum likelihood method based on the Tamura-Nei model [15] with gamma distributed with invariant sites (G+I) [14]. The robustness of the nodes was tested by 1,000 bootstrap replications. GenBank accession numbers of CCHFV genome sequences used in the phylogenetic analyses are shown in S3 Table.

## Results

### Detection of CCHFV-specific IgG in cattle

Because domestic ruminants are major vertebrate hosts for CCHFV perpetuation in disease-endemic areas, local cattle are an ideal sentinel to assess the endemicity of CCHF. To estimate CCHFV prevalence, local cattle were serologically examined by the indirect IFA using recombinant N-expressing HeLa cells [9]. This IFA has been previously used for a serological surveillance of CCHFV infection in domestic ruminants including cattle in Nigeria and showed high concordance (96%) with the results obtained by an enzyme-linked immunosorbent assay using the recombinant CCHFV N as an antigen [16]. During 2012–2015, 1,047 cattle sera were collected in 6 districts (Fig 1) and then screened for CCHFV N-specific IgG antibodies through an IFA. We found that 88 samples gave strong fluorescence signals in the CCHFV N-expressing cell-based IFA, as well as the control rabbit serum (S1 Fig). The seroprevalence was 8.4% (95% confidence interval (CI): 6.7–10.1) in total and ranged from 2.0%–17.6% depending on the sampling area (Table 1). Particularly, cattle in Mumbwa (17.6%, 95% CI: 11.1–24.1), Sesheke (14.7%, 95% CI: 9.0–20.4), and Shibuyunji (10.5%, 95% CI: 4.6–16.4) showed higher seroprevalence than those in Kazungula (2.0%, 95% CI: 0.7–5.7), Mazabuka (5.4%, 95% CI: 1.2–9.6), and Mwandi (5.8%, 95% CI: 3.5–8.1). These serologic data suggested the presence of CCHFV and its transmission to domestic cattle in Zambia. Because the virus genome was not detected in randomly selected 526 sera out of the aforementioned 1,047 sera by nested reverse transcription PCR (RT-PCR), viremic cattle seemed to be rare.

## Screening for CCHFV genomes in *Hyalomma* ticks

To directly demonstrate the presence of CCHFV in Zambia, *Hyalomma* ticks were sampled from infested cattle and genetically screened for CCHFV. During 2015–2016, 290 adult *Hyalomma* ticks (25 *H. marginatum*, 259 *H. truncatum*, and 6 *Hyalomma* spp.) were collected in 7 districts, including areas such as Mumbwa, where the CCHF seroprevalence in the cattle population was high (Fig 1 and Table 1). Total RNA of individual ticks was examined for the CCHFV N gene by nested RT-PCR, and 11 out of the 290 samples showed amplification of approximately 250 bp fragments (Table 2). Note that it cannot be ruled out that a part of positive ticks might be collected from the same cows in Chipata, Shibuyunji, and Mumbwa. Sequence analyses demonstrated that 11 amplicons had the identical nucleotide sequences and subsequent BLAST searches in GenBank revealed that the nucleotide sequence of the amplicons (GenBank accession no. LC534898–LC534908) had 100% homology with the N gene of CCHFVs found in African countries, such as South Africa, Namibia, and Sudan (S1 Table). The overall prevalence of CCHFV in the *Hyalomma* tick population was 3.8% (Table 2). Positive ticks (1 *H. marginatum* and 10 *H. truncatum*) were collected in the Choma, Chipata, Mumbwa, Shangombo, and Shibuyunji Districts (Fig 1 and Table 2). No significant correlations between the CCHFV prevalence and tick species were found by chi-square test (p-value = 0.97) (Table 3). Infectious CCHFV was not recovered from any tick homogenates.

## Phylogenetic analysis of CCHFVs

To determine longer CCHFV genome sequences, we attempted to amplify the S, M, and L genome segments from PCR-positive tick RNA samples by RT-PCR. Subsequently, 1,293 nt of the S segment sequence (nucleotide positions 176–1468; hereafter, nucleotide positions are based on the sequences of the reference strain IbAr10200) was successfully determined in 1 of the 11 PCR-positive tick samples (GenBank accession no. LC534908). Partial sequences of the M segment (nucleotide positions 25–554 (535 nt) and 4604–5118 (515 nt)) were also determined from the same tick sample (GenBank accession no. LC534909 and LC534910), while the M segment was not detected from the other 10 samples positive for the S segment. However, the L segment was not detected by RT-PCR with multiple sets of degenerate primers designed based on conserved sequences in CCHFVs (S2 Table). This CCHFV genome was detected in a *H. truncatum* tick collected in Mumbwa and designated ZT15-90. To further characterize this CCHFV detected in Zambia, the S and M genome segments were phylogenetically analyzed with CCHFV sequences retrieved from GenBank (Figs 2 and 3). As previously reported [17], the S segment-based analysis showed that CCHFVs were divided into 7 distinct genetic groups, which strongly correspond to the geographic areas where the CCHFVs were detected (Fig 2). ZT15-90 was clustered into the Africa 3 lineage and closely related to the viruses detected in Sudan and South Africa (Fig 2). The tree demonstrated that the viruses belonging to this lineage have been widely distributed in Sub-Saharan Africa, such as in Mali, Mauritania, Nigeria, South Africa, Sudan, and Uganda. In the M segment trees, CCHFVs also formed phylogenetic clusters depending on the geographic distribution, similarly to the S segment tree (Fig 3). Accordingly, most of the viruses in the Africa 3 lineage in the S segment tree clustered together and formed a unique clade in the M segment trees. However, ZT15-90 and three South African strains (SPU415/85, SPU97/85, and SPU45/88) [18] exceptionally belonged to a cluster in which the viruses in the Asia 1 and Asia 2 lineages were predominant, indicating that the M segment of ZT15-90 originated from an Asian CCHFV. This finding strongly suggests that ZT15-90 is a genetic reassortant between African and Asian CCHFVs.

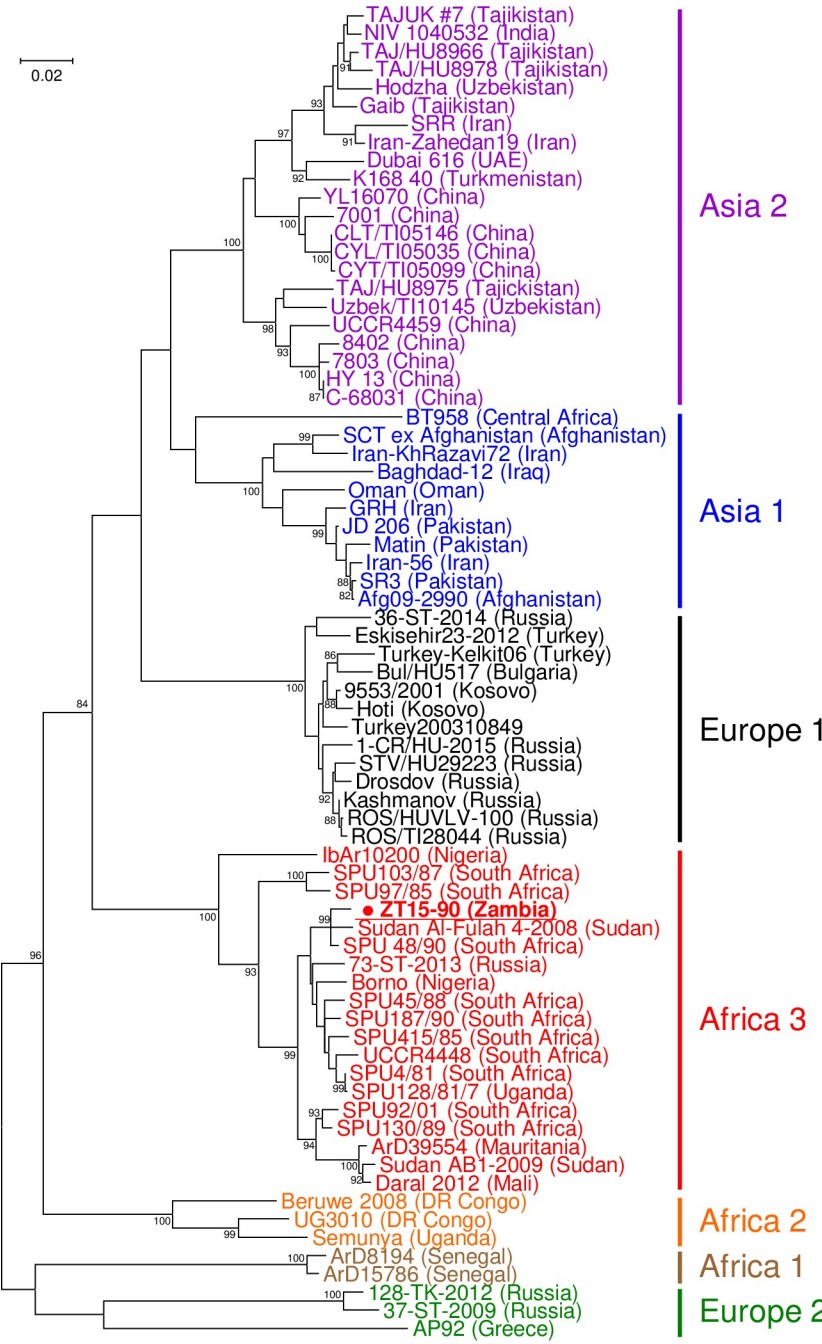

**Fig 2. A phylogenetic tree showing the genetic relationship of Crimean-Congo hemorrhagic fever viruses based on the S segment.** Genome sequences of 1,293 nt (positions 176–1468) were used to construct the tree. The evolutionary history was inferred by using the maximum likelihood method based on the Tamura-Nei model. The robustness of each node was tested by 1,000 bootstrap replicates. The percentage of tree in which the associated taxa clustered together is shown next to the branches (only 80≤). The tree is drawn to scale, with branch lengths measured in the number of substitutions per site. Countries where each strain was detected were indicated in brackets following virus names. ZT15-90 detected in the present study is shown in underlined boldface.

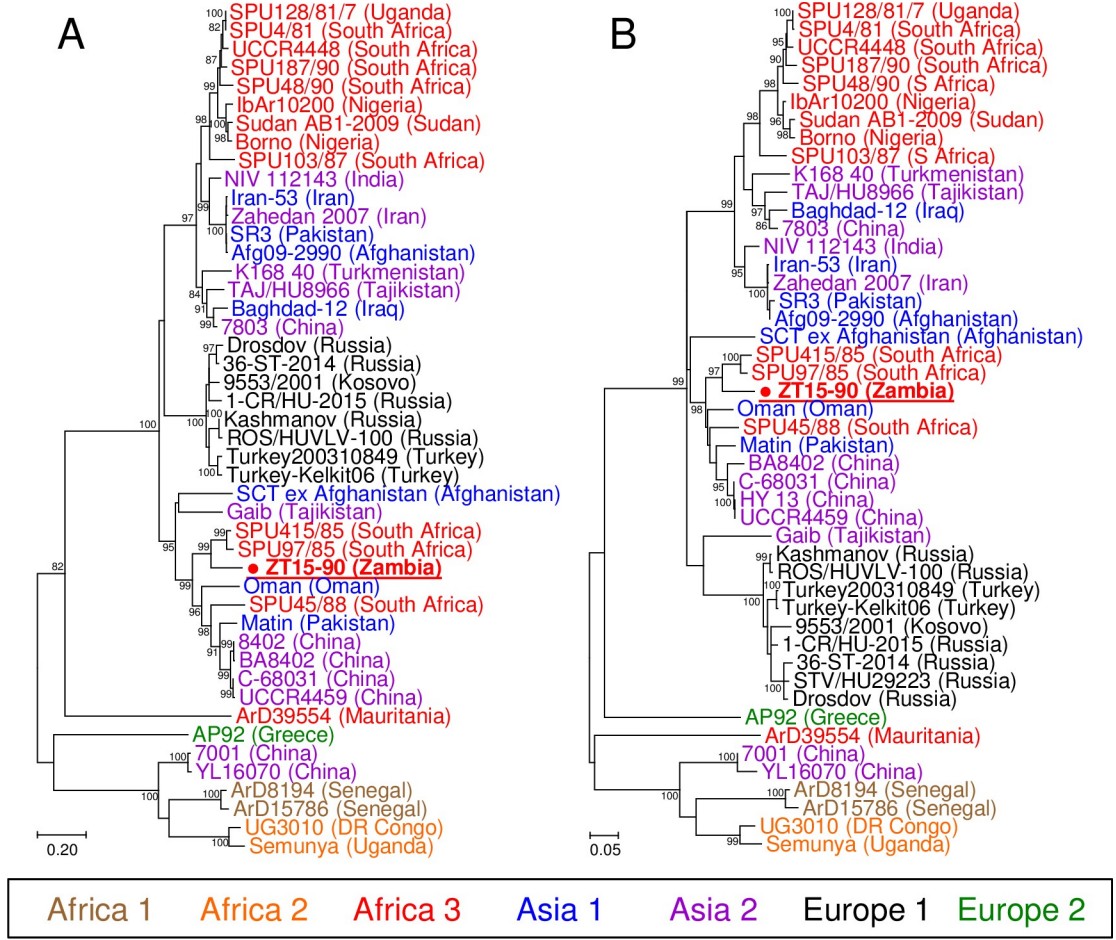

**Fig 3. Phylogenetic trees showing the genetic relationship of Crimean-Congo hemorrhagic fever viruses based on the M segment.** Genome sequences of (A) 535 (positions 25–554) and (B) 515 nt (positions 4604–5118) were used to construct the trees. Virus names are colored according to the phylogenetic groups shown in Fig 2. Countries where each strain was detected were indicated in brackets following virus names. ZT15-90 detected in the present study is shown in underlined boldface. For the method, see the legend of Fig 2.

## Discussion

CCHFV has been expanding its geographic distribution from disease endemic areas to regions previously considered CCHF free [19]. Despite the public health importance of CCHF, available epidemiologic data in Sub-Saharan Africa are quite limited. To our knowledge, the presence of CCHFV has never been empirically demonstrated in Zambia because no CCHF patients have been identified in Zambia and few epidemiologic studies on CCHF have been conducted. In the present study, CCHFV genomes were detected in *Hyalomma* ticks collected in Zambia, proving that CCHFV exists in the country. Importantly, *Hyalomma* ticks positive for the CCHFV genome were identified in 5 out of 7 sampling areas (Table 2). Serologic screening also demonstrated that cattle seropositive for CCHFV were found in all the sampling areas (Table 1). These genetic and serologic data indicate that CCHFV is widely distributed in Zambia and has been maintained in ticks and vertebrate hosts without causing an apparent outbreak in Zambia.

In this study, we detected CCHFV N-specific IgG in 88 out of 1,047 cattle serum samples (Table 1). Compared to seroprevalence in cattle in other Sub-Saharan African countries, such

as South Africa (26.5–28%) [20, 21], Zimbabwe (45%) [20], and Nigeria (24–25.7%) [16, 22], the seroprevalence in Zambia (8.4%, 95% CI: 6.7–10.1) was not remarkably high, suggesting that the virus circulation between ticks and vertebrate hosts may be also moderate in Zambia than other endemic countries. However, the higher seroprevalence in cattle in Mumbwa (17.6%, 95% CI: 11.1–24.1) and Sesheke (14.7%, 95% CI: 9.0–20.4) is likely associated with a relatively higher risk of CCHF in these districts (Table 1), because tick bites and direct contacts with infected livestock through slaughtering and farming are the major transmission modes of CCHFV. The initial symptoms of CCHF are nonspecific, including fever, headache, joint pain, and myalgia. It is also known that CCHFV causes subclinical infection in humans [23]. Although an outbreak of CCHF has never been recorded in Zambia, it is possible that sporadic CCHFV infection has been overlooked or misdiagnosed as other endemic febrile illnesses, such as malaria [5]. The present study clearly highlights that CCHF should be considered in a differential diagnosis of suspected viral hemorrhagic fever cases in Zambia to promptly contain CCHF at the beginning of potential outbreaks.

It has been reported that phylogenetic clusters of CCHFVs are highly congruent with their geographic distribution [17]. Among the 3 different African lineages, viruses in the Africa 3 lineage are most widely distributed in the African continent. ZT15-90, one of the viruses detected in Zambia, also belonged to the Africa 3 lineage and was closely related to viruses detected in South Africa and Sudan (Fig 2). In the M segment-based trees (Fig 3), ZT15-90 was grouped into a clade dominantly consisting of lineages Asia 1 and 2 CCHFVs. This result suggests that the M segment of ZT15-90 has its genetic origin in Asian CCHFVs. Thus, ZT15-90 is most likely a genetic reassortant between African and Asian CCHFVs. It should be noted that acquisition of the M segment from Asian strains has been implicated in increased pathogenicity of CCHFVs of African origins [18].

African CCHFVs with the Asian M segment have been previously reported only in South Africa and Namibia [24]. The detection of ZT15-90 in Zambia supports the notion that the genetic reassortants between African and Asian CCHFVs are more widely distributed in the southern African region. The ancestral virus that provided the M segment for ZT15-90 might have been introduced from Asia into the southern African region via cross-border animal migration or international animal trade. Indeed, CCHFV has been detected in ticks infesting migratory birds in Italy, Turkey, Greece, and Morocco, suggesting long-distance movement of CCHFVs [25–28]. According to the M segment-based trees (Fig 3), the M segments of ZT15-90 and 2 South African strains, SPU415/85 and SPU97/85, likely derived from a common Asian ancestor. However, SPU97/85 was less related to ZT15-90 and SPU415/85 in the S segment-based tree (Fig 2), suggesting that these South African viruses acquired the Asian-origin M segment at different time points. We assume that the Asian-origin M segment may be frequently exchanged among African CCHFVs through genetic reassortment. To our knowledge, CCHFV with the Africa lineage M segment has never been found outside Africa except for Spain [29]. In addition, there are no reports of African CCHFV with the Asian-origin S or L segment, whereas the viruses with the African-origin S segment have been detected in Russia and United Arab Emirates [30]. The mechanism underlying the high frequency of M segment reassortment compared to the S and L segments is totally unclear. The N and RNA-dependent RNA polymerase encoded by the S and L genome segments, respectively, might be more important factors than proteins encoded in the M segment to effectively infect endogenous vertebrate or tick hosts, thereby efficiently maintaining viral lifecycle in Africa. However, the CCHF phylogeography has not been comprehensively understood yet due to limited and geographically biased sequence data of the viruses. Further genetic and phylogenetic analyses using larger sequence datasets, including the L segment sequences, will improve our understanding of CCHFV evolution in Africa.

As the World Health Organization designated CCHF as a priority disease for research and development [31], the threat of CCHF to global public health has been increasing. Because of insufficient and biased information, the genetic diversity, phylogeography, and evolutionary history of CCHFVs are still far from fully understood. As exemplified by this study, it is conceivable that CCHFVs have already been distributed and silently circulating in ticks and vertebrate hosts in southern African countries that are currently considered CCHF free, such as Angola, Botswana, Malawi, and Mozambique. The present study underscores the strong need to implement large-scale epidemiologic studies and continuous monitoring of CCHFV infection in the southern African region.

## Supporting information

**S1 Fig. Immunofluorescence patterns of HeLa cells expressing Crimean-Congo hemorrhagic fever virus (CCHFV) nucleoprotein (N).** Expression of CCHFV N was confirmed with CCHFV N-immunized rabbit serum (A). Local cattle sera were screened for CCHFV N-specific IgG through an immunofluorescence assay. Typical fluorescence patterns of positive cells with cattle serum are shown (B).
(TIF)

**S1 Table. Sequences showing 100% homology with nucleotide sequences of partial N gene of CCHFV in Zambia in BLAST search.**
(XLSX)

**S2 Table. Primer sets used for CCHFV L genome segment detection.**
(XLSX)

**S3 Table. Accession numbers of nucleotide sequences of CCHFV S and M segments used for phylogenetic analyses.**
(XLSX)

## Acknowledgments

We thank Ladslav Moonga, Joseph Ndebe, Evans Mulenga, and Penjaninge Kapila in the School of Veterinary Medicine, University of Zambia for tick collection and Hiroko Miyamoto and Aiko Ohnuma in the Hokkaido University Research Center for Zoonosis Control for laboratory assistance. We also thank the Central Veterinary Research Institute and Veterinary Service, Ministry of Livestock and Fisheries and the Virology Laboratory, University Teaching Hospital, Zambia, for sample collection and sharing.

## Author Contributions

**Conceptualization:** Masahiro Kajihara, Jiro Arikawa, Ayato Takada, Kumiko Yoshimatsu.

**Data curation:** Masahiro Kajihara, Akina Mori-Kajihara, Yoshiki Eto, Jiro Arikawa, Ayato Takada, Kumiko Yoshimatsu.

**Formal analysis:** Masahiro Kajihara, Akina Mori-Kajihara, Yoshiki Eto, Jiro Arikawa, Kumiko Yoshimatsu.

**Funding acquisition:** Masahiro Kajihara, Hirofumi Sawa, Ayato Takada.

**Investigation:** Masahiro Kajihara, Martin Simuunza, Ngonda Saasa, George Dautu, Akina Mori-Kajihara, Yongjin Qiu, Ryo Nakao, Yoshiki Eto, Hayato Furumoto, Bernard M.

Hang'ombe, Yasuko Orba, Hirofumi Sawa, Edgar Simulundu, Jiro Arikawa, Kumiko Yoshimatsu.

**Methodology:** Masahiro Kajihara, Jiro Arikawa, Kumiko Yoshimatsu.

**Resources:** Martin Simuunza, Ngonda Saasa, George Dautu, Hayato Furumoto, Shuetsu Fukushi, Shigeru Morikawa, Masayuki Saijo, Jiro Arikawa, Swithine Kabilika, Mwaka Monze, Victor Mukonka, Aaron Mweene, Ayato Takada, Kumiko Yoshimatsu.

**Software:** Masahiro Kajihara.

**Supervision:** Jiro Arikawa, Ayato Takada, Kumiko Yoshimatsu.

**Visualization:** Masahiro Kajihara.

**Writing – original draft:** Masahiro Kajihara.

**Writing – review & editing:** Masahiro Kajihara, Martin Simuunza, Ngonda Saasa, Yongjin Qiu, Yoshiki Eto, Yasuko Orba, Hirofumi Sawa, Edgar Simulundu, Shuetsu Fukushi, Shigeru Morikawa, Masayuki Saijo, Jiro Arikawa, Ayato Takada, Kumiko Yoshimatsu.

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
