## [Decision Letter · Decision Letter 0]

22 Feb 2021

Dear Prof. Takada,

Thank you very much for submitting your manuscript "Serologic and molecular evidence for circulation of Crimean-Congo hemorrhagic fever virus in ticks and cattle in Zambia" for consideration at PLOS Neglected Tropical Diseases. As with all papers reviewed by the journal, your manuscript was reviewed by members of the editorial board and by several independent reviewers. In light of the reviews (below this email), we would like to invite the resubmission of a revised version that takes into account the reviewers' comments. 

We cannot make any decision about publication until we have seen the revised manuscript and your response to the reviewers' comments. Your revised manuscript is also likely to be sent to reviewers for further evaluation.

Sincerely,

Jonas Klingström

Associate Editor

Jeremy Camp

Deputy Editor

Reviewer's Responses to Questions

**Key Review Criteria Required for Acceptance?**

**Methods**

-Are the objectives of the study clearly articulated with a clear testable hypothesis stated?

-Is the study design appropriate to address the stated objectives?

-Is the population clearly described and appropriate for the hypothesis being tested?

-Is the sample size sufficient to ensure adequate power to address the hypothesis being tested?

-Were correct statistical analysis used to support conclusions?

-Are there concerns about ethical or regulatory requirements being met?

Reviewer #1: The design of the study was simple and appropriate to address the author’s objective. The methods used in the study were also appropriate. One request from the reviewer is to provide the primer sets (and the target position) for the L gene. Although authors described that they could not detect L gene, it should be helpful to provide that information.

Reviewer #2: The authors describe the first evidence of CCHFV in Zambia based on the detection of IgG antibodies to nucleocapsid protein in 88 cattle sera collected 2012-2015 and the presence of viral RNA in 11 Hyalomma ticks collected 2015-2016. The objective and the public health relevance are clearly described and the manuscript is well written. 

Can the authors please provide a statement in Methods confirming that the experimental protocol/protocols for sampling cattle were approved by a licensing committee (name the authorizing body and ethical permit number). If the authors feel that this is not relevant to their study, depending on the regulations in Zambia, can the authors please provide a statement in Methods explaining why there is no need for such a permit. 

The authors refer to reference 8 for the cattle samples. The paper referred to describes the collection of 942 samples in 2014. Can the authors please add information on the remaining samples or provide an additional reference describing their origin?

The authors use an in-house IF method for detection of IgG antibodies to recombinant N protein expressed in HeLa cells. As the authors present the first evidence of CCHFV in Zambia, providing more information on the specificity and sensitivity of the assay using sera from cattle, a publication describing the use of this assay with cattle sera, or confirming the results by detection of anti-glycoprotein antibodies would further strengthening the manuscript.

Reviewer #3: (No Response)

**Results**

-Does the analysis presented match the analysis plan?

-Are the results clearly and completely presented?

-Are the figures (Tables, Images) of sufficient quality for clarity?

Reviewer #1: The results presented in this study was shown clearly.

Reviewer #2: Lines 176-179, 269-276, and 273-274: The authors present and discuss an estimation of the seroprevalence in different districts, can the authors please provide the 95% CI for these estimations.

Reviewer #3: (No Response)

**Conclusions**

-Are the conclusions supported by the data presented?

-Are the limitations of analysis clearly described?

-Do the authors discuss how these data can be helpful to advance our understanding of the topic under study?

-Is public health relevance addressed?

Reviewer #1: Conclusion of the study was clear and well supported from the presented data.

Reviewer #2: The authors have drawn somewhat strong conclusions from the phylogenetic analyses which are based on a limited number of sequences. Can the authors please address these limitations in the Discussion.

Reviewer #3: Authors conclude that CCHFV has moderate circulation throughout Zambia and discuss how their data add to the field's understanding of CCHFV circulation in Africa.

**Editorial and Data Presentation Modifications?**

Reviewer #1: The reassortant between African and Asian CCHFVs is one of the exciting finding in this manuscript. Reviewer requested to author to discuss a little bit more about the reassortant. Is there any CCHFV reassortment reported of African M segment in Asia? Why only M segment is reassorted in the African CCHFV? Is it possible that L segment is also originated from Asian CCHFV in African CCHFV?

Reviewer #2: Line 96: Please clarify if the 1047 sera were collected from individual animals or not.

Lines 180-181: Please clarify if the 526 sera tested by nested RT-PCR was part of the 1047 sera tested for antibodies. In Methods, provide information regarding the treatment and RNA extraction of the sera prior to PCR analysis.

Lines 202-203: Provide the statistical test used and the p-value for determining possible correlation between prevalence and tick species. Also, in Methods add information on which software that was used to perform the analysis. 

Please add information in Methods on the BSL facility and the procedures and conditions used for the attempts to isolate CCHF from tick homogenates (described in Results, line 203).

Lines 218-219: Although it was not possible to obtain S segment full-length sequences from the other 10 RT-PCR positive ticks, did the authors try to recover partial sequences of the M segment from these ticks to confirm the result?

Line 224-237: Parts of the data presented here are/confirm previous findings from reference 16, please move these parts to the Discussion. 

Please provide GenBank accession numbers for the sequences used in the phylogenetic analysis. These accession numbers can either be added to the trees or be presented in a table in supplementary linking the names used in the trees with their respective accession number.

Reviewer #3: (No Response)

**Summary and General Comments**

Reviewer #1: Authors reported the circulation of CCHFV in Zambia for the first time. Although authors could not isolate the live virus from their samples, the data shown in here with the serologic data and the genetical data, was enough to support their conclusion. Only few requests were provided from the reviewer above to support their discussion.

Reviewer #2: (No Response)

Reviewer #3: The manuscript by Kajihara et al investigates the seroprevalence of CCHF in cattle and presence of CCHFV RNA in ticks within Zambia, a country that has not been reported as CCHFV endemic. The manuscript is well written and data well presented. Data adds to our understanding of the circulation of CCHFV within Africa, an understudied region for CCHFV. 

Minor comments:

Line 203: Authors state that no infectious virus was recovered from any tick sample, what method was used to attempt to recover infectious virus?

Table 2: Were ticks collected from individual cattle or were multiple ticks collected from the same cow? Could positive ticks have come from the same cow?

Line 134 - 148: Were no template controls included?

What is the level of movement of cattle within Zambia? Cattle movement between Zambia and Namibia or South Africa where the Asian M segment viruses have been reported? 

Figure 1: Authors should consider adding a symbol such as a star to indicate sampling sites with positive CCHFV tick or cattle samples to make it easier to understand geographically where CCHFV was detected.

PLOS authors have the option to publish the peer review history of their article (what does this mean?). If published, this will include your full peer review and any attached files.

Reviewer #1: Yes: Shuzo Urata

Reviewer #2: No

Reviewer #3: No
---

## [Decision Letter · Decision Letter 1]

7 May 2021

Dear Prof. Takada,

We are pleased to inform you that your manuscript 'Serologic and molecular evidence for circulation of Crimean-Congo hemorrhagic fever virus in ticks and cattle in Zambia' has been provisionally accepted for publication in PLOS Neglected Tropical Diseases.

Best regards,

Jonas Klingström

Associate Editor

Jeremy Camp

Deputy Editor

Reviewer's Responses to Questions

**Key Review Criteria Required for Acceptance?**

**Methods**

-Are the objectives of the study clearly articulated with a clear testable hypothesis stated?

-Is the study design appropriate to address the stated objectives?

-Is the population clearly described and appropriate for the hypothesis being tested?

-Is the sample size sufficient to ensure adequate power to address the hypothesis being tested?

-Were correct statistical analysis used to support conclusions?

-Are there concerns about ethical or regulatory requirements being met?

Reviewer #1: Methods were clear.

Reviewer #2: (No Response)

Reviewer #3: (No Response)

**Results**

-Does the analysis presented match the analysis plan?

-Are the results clearly and completely presented?

-Are the figures (Tables, Images) of sufficient quality for clarity?

Reviewer #1: Results were shown clearly.

Reviewer #2: (No Response)

Reviewer #3: (No Response)

**Conclusions**

-Are the conclusions supported by the data presented?

-Are the limitations of analysis clearly described?

-Do the authors discuss how these data can be helpful to advance our understanding of the topic under study?

-Is public health relevance addressed?

Reviewer #1: Conclusions were well supported by the results.

Reviewer #2: (No Response)

Reviewer #3: (No Response)

**Editorial and Data Presentation Modifications?**

Reviewer #1: N/A

Reviewer #2: The authors have carefully considered the reviewers comments/questions and where appropriate, made changes to the manuscript to enhance clarity.

Reviewer #3: (No Response)

**Summary and General Comments**

Reviewer #1: Well described.

Reviewer #2: (No Response)

Reviewer #3: Authors have satisfactorily addressed all my comments.

PLOS authors have the option to publish the peer review history of their article (what does this mean?). If published, this will include your full peer review and any attached files.

Reviewer #1: No

Reviewer #2: No

Reviewer #3: No

---

## [Editor Report · Acceptance letter]

27 May 2021

Dear Prof. Takada,

We are delighted to inform you that your manuscript, "Serologic and molecular evidence for circulation of Crimean-Congo hemorrhagic fever virus in ticks and cattle in Zambia," has been formally accepted for publication in PLOS Neglected Tropical Diseases.

Best regards,

Shaden Kamhawi

co-Editor-in-Chief

Paul Brindley

co-Editor-in-Chief
